# The views of public and clinician stakeholders on risk assessment tools for post-stroke dementia: a qualitative study

Eugene Tang,[1] Catherine Exley,[1] Christopher Price,[2] Blossom Stephan,[1] Louise Robinson[1]

This paper presents independent research funded by the National Institute for Health Research (NIHR).

[1]Institute of Health and Society, Newcastle University, Newcastle, UK
[2]Institute of Neuroscience, Stroke Research Group, Newcastle University, Newcastle, UK

**Correspondence to**
Dr Eugene Tang;
e.y.h.tang@newcastle.ac.uk

## ABSTRACT

**Objective** Stroke-survivors are at increased risk of future dementia. Assessment to identify those at high risk of developing a disease using predictive scores has been utilised in different areas of medicine. A number of risk assessment scores for dementia have been developed but none has been recommended for use clinically. The aim of this qualitative study was to assess the acceptability and feasibility of using a risk assessment tool to predict post-stroke dementia.

**Design** Qualitative semi-structured interviews were conducted and analysed thematically. The patients and carers were offered interviews at around 6 (baseline) and 12 (follow-up) months post-stroke; clinicians were interviewed once.

**Setting** The study was conducted in the North-East of England with stroke patients, family carers and healthcare professionals in primary and secondary care.

**Participants** Thirty-nine interviews were conducted (17 clinicians and 15 stroke patients and their carers at baseline. Twelve stroke patients and their carers were interviewed at follow-up, some interviews were conducted in pairs).

**Results** Barriers and facilitators to risk assessment were discussed. For the patients and carers the focus for facilitators were based on the outcomes of risk assessment for example assistance with preparation, diagnosis and for reassurance. For clinicians, facilitators were focused on the process that is, familiarity in primary care, resource availability in secondary care and collaborative care. For barriers, both groups focused on the outcome including for example, the anxiety generated from a potential diagnosis of dementia. For the patients/carers a further barrier included concerns about how it may affect their recovery. For clinicians there were concerns about limited interventions and how it would be different from standard care.

**Conclusions** Risk assessment for dementia post-stroke presents challenges given the ramifications of a potential diagnosis of dementia. Attention needs to be given to how information is communicated and strategies developed to support the patients and carers if risk assessment is used.

## INTRODUCTION

There is currently no cure for dementia and it is estimated that the worldwide economic

---

### Strengths and limitations of this study

► To the best of our knowledge this is the first qualitative study to examine critically the views of stroke patients and their family carers and clinicians about the acceptability and feasibility of a risk assessment approach to assist in earlier identification of post-stroke dementia.

► Understanding stakeholder views on risk assessment for dementia can help inform future strategies if risk assessment for dementia is used to assist with earlier diagnosis.

► The patient participants came from one area of England who were able to attend hospital outpatient departments and so may not represent the views and experiences of those with more severe post-stroke sequelae.

► Clinician participants came from one area of England and so may not represent the views of other service models in other regions of the UK.

► It is recognised that clinicians tended to be more familiar with the process of risk assessment and could elaborate further on the process involved.

---

burden will rise to US$2 trillion by 2030.[1] It has been suggested that the most powerful way to affect costs is by reducing the numbers of people who develop the illness. This may be facilitated by prediction of individual risk for the disease. Stroke is associated with an increased risk of dementia and cognitive impairment.[2–4] A recent meta-analysis found that stroke is a strong independent risk factor for dementia.[5] Stroke incidence and numbers of stroke-survivors are likely to increase due to simultaneous ageing populations and declining stroke mortality rates.[6] Given that the incidence of dementia increases exponentially with age,[1 7] this will mean that post-stroke dementia will also become increasingly prevalent. It will therefore be important to identify those at greatest risk of developing dementia following stroke in order to

implement strategies to reduce risk. In general, strategies to reduce risk of dementia may include management of cardiovascular risk factors for example, smoking, diabetes as well as regular physical activity.[8]

Risk prediction models for dementia to identify those at higher risk have been developed in whole populations[9 10] with some models specifically developed to predict cognitive impairment and dementia in stroke populations.[11–14] These stroke-specific models predict dementia or cognitive impairment over a relatively short time period (up to 18 months[14]). In spite of the expanding research in this field, none of the dementia risk prediction tools have been clinically implemented. Further, no studies have assessed the feasibility or acceptability of implementing such a strategy in a stroke population. Although risk models are currently used in everyday clinical practice in other branches of medicine, in particular prevention of cardiovascular[15] and cerebrovascular[16] disease, it is unclear how clinicians would feel about using a similar strategy to predict dementia, particularly given the stigma surrounding the diagnosis and perceived limited interventions and increased awareness of cognitive difficulties that the patients and carers may have following stroke. Further, no studies have evaluated whether using risk assessment tools for dementia would be acceptable to stroke patients themselves.

This paper presents findings from a qualitative study conducted with the patients, carers and clinicians, which, in part, sought to critically examine their views about the acceptability and feasibility of using risk prediction models in post-stroke care to identify those at greatest risk of future dementia.

## METHODS
### Patient and public involvement
Patients and members of the public have been involved in the development of this study from the beginning of the proposal. A participant advisory group also oversees the work conducted and annual face-to-face meetings are held to inform them of the study findings. The participant advisory group consists of members from a stroke research patient and carer panel, an organisation aimed at capturing public views about research and from a dementia and neurodegeneration specialty patient and public involvement group. The same group reviewed the study materials to ensure suitability particularly for stroke-survivors and their family carers.

### Ethical approval
The study was conducted in the North East of England. Participants provided informed written consent prior to the interview.

### Patient and carer sampling
Patients and carers were purposively sampled from stroke clinics that is, to ensure a mix of genders and a range carers were recruited. As part of routine clinical practice in UK stroke services, all stroke-survivors are invited to a specialist review at 6 months after the event which includes a general enquiry about memory concerns.[17] If the patient reported any subjective memory concerns at the clinic and was over the age of 60 and were able to communicate effectively in English, the stroke specialist nurse would provide further study information. Family carers were also recruited if they were involved in the stroke-survivor's care, for example, if they attended the clinic appointment with them. If potential participants were interested in taking part in the study, their details were passed onto the research team. On receipt of this information one researcher (EYHT) would make contact with the patient or carer. He would provide detailed information and an opportunity to ask questions about the study. Following their agreement to participate in the study, participants were asked to take part in an interview immediately following their 6 month review and/or around 6 months later.

### Clinician sampling
General practitioners (GPs) and secondary care clinicians (eg, stroke consultants, specialist nurses, physiotherapists and occupational therapists) in the North East of England were contacted to participate in the study. Participants were given an opportunity to ask further questions. Clinicians were purposively sampled to ensure that a broad range of care professionals in both primary and secondary care were recruited.

### Data collection
Interviews were conducted between April 2016 and August 2017 by one researcher (EYHT) who is a medical doctor. The topic guide was initially derived from relevant literature and expert clinical views within the research team. It was designed to be iterative to enable any topics, which had not been previously identified, to be pursued in subsequent interviews. Face-to-face semi-structured interviews were conducted with all but one participant (clinician) who had a telephone interview. The patient and family carer were interviewed individually or in pairs as requested by participants. Clinicians were interviewed individually. The part of the interviews focussing on risk assessment asked participants for their views on using risk assessment to help identify stroke-survivors who are most at risk of dementia in the future. They were also asked about the benefits and problems associated with the delivery of this assessment (eg, who and where it should be carried out), what variables could be used and how best to manage the outcome if individuals were found to be at high or low risk. At follow-up interviews, the patient and carer participants were asked to elaborate again on their views of a risk assessment process. Alongside this, the interviews also sought the views of stakeholders on the care experience of post-stroke individuals with memory problems from clinicians, patients and carers. The interviews also looked to understand the impact of post-stroke memory problems on the patients and carers. These views

on care experience from clinicians[18] and the patients and carers[19] have been reported elsewhere. The impact of post-stroke memory problems on the patients and carers will be reported separately. This paper reports the views of clinicians, patients and carers on risk assessment only. The process of risk assessment was described to participants. This was further emphasised with examples of published tools in order to highlight examples of variables used to ensure participant understanding of the process. Informed written consent was obtained from all participants prior to the interview commencing. All interviews were audio-recorded and then transcribed verbatim. To protect participant anonymity, unique identifiers were used throughout the process with identifiable personal data removed.

## Data analysis

Interview data was analysed using thematic analysis[20] following the principles of the constant comparative method,[21] an iterative approach which allows for issues raised in earlier interviews to be explored subsequently. Data analysis was both deductive and inductive in that we applied learning from previous research and compared with our own data as well as inductively deriving new themes from our data. We ceased data collection when the researcher felt that data saturation occurred. This was defined as being when a full understanding of the participant's perspective[22] and also 'informational redundancy' had been reached.[23] One researcher (EYHT) familiarised himself with the dataset and subsequently coded the transcripts line-by-line. Initially, a small subset of transcripts were analysed to identify initial themes and these were discussed between CE and EYHT. Data collection and analysis was iterative and as interviews progressed, further analysis led to new themes emerging and refinement of existing themes and subthemes, which were subsequently grouped into broad categories to facilitate interpretation. The wider team (EYHT, CE, LR, BS and CP) discussed and agreed on the final categories which are presented below. For the patient and carer interviews, where follow-up interview data was also obtained, these were analysed as separate interviews to assess for any change in views over time. Data analysis continued after fieldwork had ceased. There was particular focus to understand what was important to the patients, carers and clinicians. Data analysis was facilitated by a data handing software package (NVivo V.11). The paper conforms to the standards for reporting qualitative research checklist[24] (please see online supplementary table 1).

## RESULTS

In total, 30 baseline (6 month) interviews were conducted, analysed and compared including: 15 patient and carer interviews (see table 1) and 17 primary and secondary care clinician interviews (see table 2). Two pairs of participants were interviewed together at baseline. Eight stroke-survivors and four carers agreed to a further

**Table 1** Interview participants (patients and carers)

| Unique identifier (patients and carers) | Role | Gender | Age | Follow-up interview conducted |
|---|---|---|---|---|
| P1 | Stroke-survivor | Female | 80 | No |
| P2 | Stroke-survivor | Female | 76 | Yes |
| P3 | Stroke-survivor | Female | 72 | Yes |
| P4 | Stroke-survivor | Male | 75 | Yes |
| P5 | Stroke-survivor | Male | 80 | Yes |
| P6 | Stroke-survivor | Male | 74 | Yes |
| P7 | Stroke-survivor | Female | 73 | Yes |
| P8 | Stroke-survivor | Female | 82 | Yes |
| P9 | Stroke-survivor | Male | 84 | No |
| P10 | Stroke-survivor | Male | 79 | Yes |
| C1 | Carer of P1 (husband) | Male | 79 | No |
| C2 | Carer of P4 (wife) | Female | 79 | Yes |
| C3 | Carer of P5 (daughter) | Female | 57 | Yes |
| C4 | Carer of P6 (wife) | Female | 71 | Yes |
| C5 | Carer of P8 (daughter) | Female | 60 | Yes |

follow-up interview 6 months later with nine interviews completed. Three pairs of participants were interviewed together at follow-up. One stroke-survivor declined further follow-up and another stroke-survivor and carer were not followed up due to medical reasons. The data from this study suggest that in terms of risk assessment facilitators and barriers exist to implementation. Whereas the patient facilitators focused on the outcome of the risk assessment, clinicians focused more on the process of risk

**Table 2** Interview participants (clinicians)

| Unique identifier (clinicians) | Role | Gender |
|---|---|---|
| SC1 | Stroke consultant | Female |
| SC2 | Stroke specialist nurse | Female |
| SC3 | Stroke consultant | Female |
| SC4 | Stroke consultant | Male |
| SC5 | Stroke specialist nurse | Female |
| SC6 | Stroke physiotherapist (rehabilitation) | Female |
| SC7 | Stroke physiotherapist (acute care) | Female |
| SC8 | Stroke occupational therapist (acute care) | Male |
| SC9 | Stroke occupational therapist (rehabilitation) | Female |
| PC1 | General practitioner with specialist interest in dementia | Male |
| PC2 | General practitioner | Male |
| PC3 | General practitioner | Female |
| PC4 | Nurse practitioner in primary care | Female |
| PC5 | General practitioner | Female |
| PC6 | Practice nurse | Female |
| PC7 | Nurse practitioner in primary care | Female |
| PC8 | General practitioner | Female |

assessment for facilitators. Both groups discussed some potential barriers associated with risk assessment focussing on the outcome.

### Patient and carer views: facilitators to risk assessment focuses on the outcome of assessment

When stroke-survivors and carers discussed the concept of risk assessment, the overarching theme was that an assessment outcome was what was important, irrespective of the process and clinicians involved. Participants focused on several areas of why the outcome was important to them.

#### For preparation

Some stroke-survivors were generally positive about receiving a risk assessment for dementia. One stroke-survivor acknowledged that a diagnosis was something that could enable individuals to prepare themselves both at baseline and subsequently at follow-up interview:

> It's the same as knowing and not knowing, if you know that something is approaching. Not everybody is the same with the problem. You might be able to deal with it in a different way or the person supporting you, the nurse or whoever, might be able to find a different way or a more positive way of managing it. (P6, male, stroke-survivor at follow-up interview)

Similarly, for carers, there was the emphasis on what could be done following the assessment. One carer emphasised the importance of looking after the whole person, and, how earlier recognition of a potential dementia diagnosis could ensure strategies were in place to help the individual:

> But I think, if you look at the whole thing of this care of this person, if we knew earlier that you know the chances are that your memory is going to get bad and you are going to go into dementia or whatever, then we can start thinking, 'Right, well let's prop it up, let's think of ways in helping your memory as it is, to maintain the level it is before you've got no choice, it's going to get worse.' You know, maintaining what you've got and different ways of maintaining it, I think that would help. (C5, female carer (daughter) of stroke-survivor)

#### For timely diagnosis

For some stroke-survivors it did not matter who was performing the risk assessment for dementia or where it was undertaken. What was important was that the diagnosis was reached at the right time:

> I wouldn't say it matters, as long as it's diagnosed at the right time. (P5, male stroke-survivor)

When discussing who should perform the risk assessment, carer participants felt that primary care and the community were regarded as being optimal because of the existing GP-patient relationship. This is because the GP has an overall view of the individual's care:

> I think if you've got a good relationship with your GP I think it should be that, it should be them. Yeah, because you know you trust them you build up a relationship with them so I think that probably, for me that would be the one. (C4, female carer of stroke-survivor)

#### For reassurance

When stroke-survivor participants were asked about a structured risk assessment process, a further participant reported that the outcome could also ensure some reassurance, either that their symptoms were not related to a dementia diagnosis or that a diagnosis of dementia would be accompanied by support:

> I think it's reassurance a lot of reassurance with people. You have to give them that to tell them, that 'We are there with you. We're going to be helping you.' And that's you know, I think that's a good thing. (P2, female stroke-survivor)

### Patient and carer views: barriers to risk assessment focuses on the outcome of assessment

#### Anxiety around a potential diagnosis of dementia

Some carers commented on how the outcome from risk assessment could generate worry and anxiety because of the potential diagnosis of dementia:

> To be honest, I don't know if it would help somebody saying, 'You're like this, you're upset because you're like this now, but we actually think you're going to get much worse.' Do you know what I mean? (C3, female carer (daughter) of stroke-survivor)

This person's opinion did not change when she was followed up 6 months later. The participant's focus was again on worrying about what could develop and how not knowing about one's risk would actually be more preferable:

> If you could find out and then say, 'Right, we've got this medication, or something, that can help you,' maybe. But if they're just going to tell you, and then you've got this hanging over your head, and you're thinking, 'When is it going to start?' and then you'd be thinking you'd forget something and you'd think, 'Oh, that's it, it's coming', which it would be quite normal if you hadn't had that diagnosis, you'd think, 'Well I just forgot something, everybody does that. (C3, female carer (daughter) of stroke-survivor at follow-up interview)

However, one carer felt that despite the worry a potential diagnosis may generate, the benefit of this would be to find strategies to maintain cognitive function:

> I think if you had earlier diagnosis, then you would be sort of prepared before things got difficult to handle, or before problems arise, that would be a very good thing. The disadvantages as you say, alarming the

carers or the patients themselves, 'I'm going to lose my mind.' Because, particularly in the older generation, that's a big worry to them. It is a big worry, it's a big worry to all of us, but to older people particularly. (C5, female carer (daughter) of stroke-survivor)

## Concerns about how it may affect their recovery

Not all stroke-survivors were as keen to engage in risk assessment, as there was emphasis on how this may affect them psychologically particularly when their physical deficits had recovered enough to allow them to return to a more usual routine. Therefore, although diagnosis was felt to be important, whether an individual would like to know was also dependent on their subsequent post-stroke recovery:

That's difficult you know because I mean if you have an early diagnosis you know and say, well 'It's going to happen' you know but at the moment now I seem to be progressing through, I'm driving now, you know I'm going back to meetings and whatever. I wonder whether an early diagnosis would restrict that. (P4, male stroke-survivor)

This was particularly evident when the patients were followed up 6 months later. One participant had actually changed her view over time. Although she had initially felt positive about the process, she then changed her mind when questioned on the same process at her follow-up interview:

I think my thinking has gone the other way for knowing about that. I think it's sad. I think it's a sad thing. I really do, I think it's really sad that for people to know that they're going to be at high risk, it's a sad thing for it to happen to people, and I don't think I'd want to be one of the sad people. I think I'd just want to be, potter along and that's it. (P2, female, stroke-survivor at follow-up interview)

At follow-up interviews participants also felt that risk assessment should be an individual choice because of the ramifications of the assessment outcome that is, a potential diagnosis of dementia. Although clinicians may deem it to be helpful, the choice to undergo risk assessment needs to be a weighed up, which should negate any calls for it to be made a universally applied process:

I think, medically speaking, yes. On the other hand, does it give people things to worry about that they wouldn't have worried about if you hadn't done the tests? So, I think it depends really on your personal point of view. Do you want to be, you see I would look on the test as saying, well you're at a low, you've got a low risk so that's great but then if it turned out you'd got a high risk are you going to be more worried and less happy than you were before. It's hard to really balance it, isn't it? (P3, female, stroke-survivor at follow-up interview)

## Clinician views: facilitators to risk assessment focusses on the process

Clinicians discussed facilitators to risk assessment in terms of how the process may affect the individual and also how the process could be implemented in the future.

When discussing how to implement this process, both primary and secondary care specialists discussed the advantages associated with hosting this process within their own individual teams.

### Process familiarity in primary care

For primary care, it was about the fact that risk assessment was already a familiar process but that it needed to be individualised:

I think it's a good tool. We're quite good at using tools, aren't we, but there's always going to be exceptions to the rules and you've got to individualise what you do with it … But sometimes using a score or a tool is a way into a service. (PC4, nurse practitioner in primary care)

It was also recognised by one GP that although there is familiarity with risk assessment in primary care, there needs to be caution that the system is not overwhelmed with such tools:

I do quite like risk profiling. I think we went a little bit crazy with the risk profiling. And there feels to be a lot of competing risk profiling tools, that we're getting a little bit inundated with at the moment … So I think anything like this, I love, if it can be incorporated and brought on to an individual and needs level - so you can think about caring, identifying risk and needs for an individual - would feel great for me. (PC2, general practitioner)

### Secondary care provides specialist input

Stroke care clinicians discussed the facilitators of risk assessment within a specialist setting. This was based on the fact that they felt a responsibility to ensure that post-stroke sequelae are followed up in their specialist services due to the multidisciplinary element of their standard practice and easier access to services. This was particularly important to ensure information could also be given to the patients at a time when they may need it the most:

I think the 6 month review tends to be a period of time when the patient's acute side, acute phase of their care has kind of been established, and this is probably the time when they start to recognise problems. And I think it should be within a stroke MDT (multidisciplinary team), not so much focused on by GP's, as such. (SC2, stroke specialist nurse)

Well, you need the right support. You need people that actually understand stroke. So I think it would have to be delivered by stroke healthcare professionals. And I think you get so much information when you're initially an inpatient, I think maybe

that's not the best place to do it … Yeah, it's a big thing to be told that you might develop dementia in a few years' time, so you need psychologists kind of available for if someone needs counselling as a result of that finding. I think it's tricky. (SC6, stroke physiotherapist)

### Collaborative care

Primary care clinicians commented that there may be a place for both primary and secondary care to work together in identifying those at risk.

I think primary care would be a completely reasonable place to do that. I guess it's a conversation that could start at diagnosis, at discharge from hospital, like actually, we know that people who have had a stroke are at higher risk of having dementia, these are the things to be aware of, and you know to start that discussion (PC8, general practitioner)

Primary and secondary care clinicians felt that such a shared care pathway needed to be formalised to reduce the risk of individuals falling into gaps in care:

… even if it was picked up in secondary care it's still going to be primary care where most of the management is occurring. So I think it being identified at the 6 month follow-up, but then there being a formal sort of mechanism, in which primary care pick it up and process it, would be fine. (PC3, general practitioner)

I don't mind where work is done, provided that it is done in a structured and standardised way. If that be, if that can be in primary care that is really good, because that is the long-term follow-up, long-term support, integrating the community … just as long as it can be delivered in a systematic way, and people don't fall through gaps or get inconsistent care. (SC3, stroke consultant)

Further, the process of communication between primary and secondary care could also be used in the diagnostic process. It was felt that repeated assessments could help facilitate diagnosis by identifying trends in symptoms:

You can measure a trend, can't you, if you're using something and measuring something, you can look at a trend. So if its, depends on the type of tool, I guess. But if you did it at you know at the 6 months review date and then we did it subsequently a year later in primary care, you would see any changes or decline or improvement. So it's a way of, it's a way of monitoring a trend on how they're doing, I guess. So I don't, I don't see any reason why it couldn't be done in both and used across both. I don't think we use enough across both. (PC4, nurse practitioner in primary care)

### Clinician views: barriers to risk assessment focusses on the outcome

#### Limited interventions available

Similar to the perspectives of carers, clinicians recognised the anxiety that a risk assessment process might generate and felt that it should be a personal choice to undertake an assessment because of the perceived lack of intervention:

Yeah, I think I would, I would have degree of anxiety, especially given that the measures that we're putting in place are … that we could put in place are largely supportive rather than preventative … I would be less confident that I could be giving my patient advice to say, Well, if we do this, and we do this, and if we do this and you do that then that might move you into an even smaller risk group. (PC3, general practitioner)

Outside research trials, I'm not convinced that there is a definitive value in doing that yet. You know if we get really overwhelming evidence that it's amenable to intervention so you know there's all the theory about blood pressure, and statins, and all the rest of that, but my reading of the evidence on all of that at the moment is that the jury is out whether it makes a difference to cognitive function. So yeah, I'm not convinced that identifying risk, unless you've got a something you can do about it, is actually sensible. (SC4, stroke consultant)

#### Anxiety around a potential diagnosis of dementia

In recognising the anxiety that this process may generate, one clinician also commented on the fact that the patients may not be willing to engage in conversation over the subject of dementia and care should be taken when discussing a potential diagnosis of dementia.

I think it's good if we tell them that we're looking through and saying, 'Look, you know there could be a problem here.' But for every single patient, again, because it's quite a still a – not a taboo subject – but it's still not something that people want to talk about … I don't know whether it would be used on every single 'per', you know what I mean, like, everybody. (SC5, stroke specialist nurse)

#### No change from standard practice

The majority of clinical participants wanted to know, not only what the outcome of the risk assessment would be, but also the resulting care the patient would receive. As part of current routine clinical care, all stroke-survivors are offered annual reviews in order to ensure their vascular risk factors for example, blood pressure and cholesterol are well controlled. In terms of reducing risk, one primary care physician expressed concerns as to what the benefit would be to the individual if risk factor modification was already in place anyway, particularly with regards to the emotive side of a potential dementia diagnosis. A secondary care specialist questioned the value when there was seemingly limited interventions that

could be implemented besides managing their cardiovascular risk:

> I guess you've got to be very clear about what it is that you're going to be doing differently for them. So I can see the value if you use a tool for kind of primary prevention, then you're kind of selecting a group of patients out to do something particular with, but I just wonder what would be different about what you do with a risk assessment tool for people who have already had a stroke, when really you know already that it is all about managing their cardiovascular risk so I'm not sure that you would be doing anything different for them. (PC8, general practitioner)

> Many people will not know of the association between dementia and stroke and many people would not want to know if they were at risk of dementia and again, if you're identifying somebody at risk of a condition that you can't do anything about, what's the right stage to, to do that? However, many of the things you need to do in terms of people being at risk of dementia are the same of the general cardiovascular. So, I'm not sure that there is anything additional that needs to be done about reducing people's risk for dementia over and above general cardiovascular risk. (SC3, stroke consultant)

## DISCUSSION
### Main findings
This is the first study to explore key stakeholders' - stroke-survivors, family carers and primary and secondary care clinicians - views on the use of a risk assessment process to predict future dementia in stroke-survivors. It is clear that some of the participants interviewed believed that risk assessment could be of clinical use, but raised concerns about it being mandatory. Clinicians highlighted both the benefits of collaborative and individual (ie, primary or secondary) care if dementia risk assessment for stroke-survivors was to be implemented.

Clinician facilitators suggest benefits in either primary or secondary care settings, but also in a collaborative model of care between the two. This latter finding echoes recommendations from the UK Intercollegiate Stroke Working Party for a collaborative care model, linking community and specialist care, with the aim of integrated long-term follow-up for those presenting neuropsychological problems.[17] Although both primary and secondary care clinicians could see the benefits of carrying this assessment in their own specialties, some of the patients and carers in this study valued their relationship with their GP. Further, primary care clinicians themselves are familiar with the process of risk assessment. A survey of primary care physician trainees found that they were also keen to implement a dementia risk assessment strategy to assist in earlier identification.[25] However, potential barriers have been identified in previous studies, such as system-related factors (lack of support, time constraints)[26 27] and training

in dementia,[27] which would need to be addressed. Risk assessment is an objective process requiring specific individual variables for example, age, gender and education. Such data is readily available in primary care in many countries where electronic medical record systems are in place. Further, GPs are already asked to assess cardiovascular risk as part of routine clinical care.[28] However, some GPs themselves do not like using risk assessment tools particularly as the tools do not provide the support needed in communication.[29] Training in communicating the risk assessment process, particularly in the context of dementia, would be required if this were to be implemented in clinical practice. Further, some models, particularly those developed in stroke populations[11] may also include variables such as complex imaging data, which will only be available in secondary care and may be difficult to obtain even in specialist settings. If risk assessment were to be conducted in primary care, then the risk assessment models utilising data which can be accessed in primary care, needs to be externally validated in stroke populations to assess their accuracy.

Clinician participants were concerned about whether risk assessment would actually change standard practice. In a stroke population, it is unclear whether identifying those at risk would achieve any additional benefit from a risk factor modification point of view. This is because stroke-survivors already receive annual community follow-up with particular focus on vascular risk factor modification. However, current evidence suggests that development of post-stroke dementia is more than just about vascular risk and would require a different approach for example, psychological support, cognitive preservation strategies and additional resources. Results from several trials, assessing whether vascular-based interventions can reduce dementia risk, have been largely disappointing.[30 31] These results suggest that perhaps an individual's risk of post-stroke cognitive impairment and dementia includes risk factors beyond vascular risk. Inflammation following a stroke seems to have both positive and negative effects and whether lowering inflammation can prevent post-stroke dementia will need to be addressed in future trials.[32]

Currently population screening for dementia is not recommended due to a lack of evidence evaluating risks and benefits,[33] despite positive views from older adults.[34] Risk assessment can target high-risk groups rather than the general population. Recent evidence has found a decline in age-specific incidence of dementia, particularly in high-income countries, suggesting that rising levels of education and modifying cardiovascular risk may have driven a decline in dementia risk.[35 36] Indeed, the importance of modifiable risk factor reduction for dementia was reported in the World Alzheimer Report (2014)[37] and around a third of Alzheimer's disease cases worldwide might be attributable to modifiable risk factors.[38] Risk assessment tools use these modifiable risk factors to predict risk. Similar to other branches of medicine where risk assessment is used to predict risk of a future illness,

it would be hoped that this approach could reduce one's risk of future dementia. Stroke affects more than 100 000 people in the UK per year,[39] creating a large population with cognitive deficits and/or at high risk of future decline who may benefit from risk assessment for dementia. However, participant groups in this study, particularly clinicians, reported that given the potential ramifications of risk assessment, individuals should be given the choice of whether to undergo assessment. Some stroke-survivors were positive about such an approach, but agreed that it should be up to the individual and the family rather than applied universally. Participants in this study recognised the anxiety this process could generate particularly when the perceived possible interventions for dementia are limited. The National Institute for Health and Care Excellence have recently updated their guidance and have concluded that case finding should only be conducted as part of a clinical trial, which also provides an intervention.[40] Therefore, careful discussion needs to be adopted with the the patient and their carers before undertaking such a process in any setting. In the context of the dementia diagnostic journey, transition from living with an undiagnosed memory problem to being diagnosed with a dementia illness is underpinned by uncertainty.[41] Although risk assessment certainly does not provide any certainty for a dementia illness, the discussions and objective evaluation using the tools may help the individuals process their current condition and assist in the preparation for a potential diagnosis of dementia. Preparation was mentioned by participants in this study as a facilitator for risk assessment.

### Clinical implications

Case finding for dementia involves actively assessing individuals at risk of a future dementia illness, which at present is only recommended in clinical trial settings due to a lack of post-assessment intervention.[42] Once a suitable intervention is found however, the views of those conducting the assessment and the recipients of such an assessment will need to be assessed. Similarly there will be challenges with regards to assessment of capacity when performing risk assessment for this at-risk population. It is also important to note that GPs find communicating the diagnosis of dementia difficult.[43] Although risk assessment is not providing a diagnosis of dementia, careful consideration will be required in training health professionals in communicating the concept of risk for a disease such as dementia. From this study we have identified the priorities according to each stakeholder group which would need to be addressed prior to clinical implementation in the future.

### Limitations

The participants in this study came from one area of England and were Caucasian. The patient participants were also well enough to attend outpatient assessment clinics. Future studies could look to explore views in other populations including views from minority ethnic groups,

the patients with more severe stroke-related impairments and different service models. Due to familiarity, it is recognised that clinicians expanded more around the risk assessment process. Despite this being the case, the patients and carers were given the opportunity to understand the concept of risk assessment as part of the interview process, but the emphasis on a need for a diagnosis and good care was what was important for them. Participants were also aware that the interviewer was also a primary care clinician, which may have the potential to introduce bias into participant responses. This is because a clinician interviewer may be viewed as an expert and judge in clinical decision making and moral judgements made.[44] On the other hand interviews tend to be broader in scope and richer in data when conducted by a clinician researcher.[44] Further, both clinical and non-clinical members contributed to the analysis of the data to minimise the effect this may have had.

## CONCLUSIONS AND FUTURE RESEARCH

Timely recognition of those at risk of dementia is crucial to enable individuals early access treatment and support. Although dementia screening after stroke is not yet advocated on preventative grounds, assessing risk has some potential benefits for individuals who make an informed choice to participate. There would need to be better cohesiveness of communication between primary and secondary care, with more support placed in the community. Further, it should be recognised that if risk assessment were to be incorporated into clinical practice, this will potentially place additional burdens on a dementia diagnostic service, which is already overstretched. Next steps are to identify which tool to use, how best to manage those who are deemed high-risk individuals and whether there are any interventions, which can reduce their risk. Future studies will need to look specifically at what factors put a stroke-survivor at risk that could be potentially modified and also whether there are specific interventions suitable to a post-stroke population to reduce risk.

**Twitter** @eugeneyhtang

**Acknowledgements** The authors would like to thank the participant advisory group for their advice on the study materials used.

**Contributors** ET conceived the framework for this study. ET collected, analysed and interpreted the data. ET prepared the manuscript for submission. CE helped to conceive the framework for this study and assisted with the analysis of the data and contributed to the drafting of the manuscript. CE also critically reviewed and edited the manuscript. CP helped to conceive the framework for this study, assisted with the analysis of the data and critically reviewed and edited the manuscript. BS helped to conceive the framework for this study, assisted with the analysis of the data and critically reviewed and edited the manuscript. LR helped to conceive the framework for this study, assisted with the analysis of the data and critically reviewed and edited the manuscript.

**Funding** Eugene Tang is supported by a NIHR Doctoral Research Fellowship (DRF-2015-08-006). Louise Robinson is supported by a National Institute for Health Research professorship (NIHR-RP-011-043) and a NIHR Senior Investigator award (NF-SI-0616-10054).

**Disclaimer** The views expressed are those of the author(s) and not necessarily those of the NHS, the NIHR or the Department of Health and Social Care.

**Competing interests** LR reports grants from NIHR Professorship award, grants from NIHR Senior Investigator award, outside the submitted work.

**Patient consent for publication** Obtained.

**Ethics approval** Ethical Approval was obtained from the London – Hampstead Research Ethics Committee (reference 16/LO/0133).

**Provenance and peer review** Not commissioned; externally peer reviewed.

**Data sharing statement** No further data will be made available.

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
