## [Reviewer comments · BMJ Open]

ARTICLE DETAILS

TITLE (PROVISIONAL)	The Views of Public and Clinician Stakeholders on Risk Assessment Tools for Post-Stroke Dementia: A Qualitative Study
AUTHORS	Tang, Eugene; Exley, Catherine; Price, Christopher; Stephan, Blossom; Robinson, Louise

VERSION 1 - REVIEW

REVIEWER	Dr Olivia Hewitt The Oxford Institute of Clinical Psychology Training, University of Oxford, UK
REVIEW RETURNED	16-Aug-2018

GENERAL COMMENTS	Views of Public and Clinician Stakeholders on Risk Assessment Tools for Post-Stroke Dementia: A Qualitative Study Thank you for submitting this interesting paper for review. I hope you will find the following comments helpful. Formatting • The size and format of the text should be consistent throughout the paper. Abstract • It's not clear what 'using a risk model to predict post stroke dementia' actually means in this context.• "Qualitative semi-structured interviews were conducted with thematic analysis." Please change to clarify that the interviews were analysed (not conducted) using thematic analysis. How long post stroke were the interviews conducted? Were they conducted individually (one to one)?• Section on 'Participants' needs re wording as is not currently grammatically correct.• Thirty-nine (17 clinicians and 15 stroke patients and their carers at baseline) 17 plus 15 isn't 39 – could you be explicit about how many patients are seen, how many staff and how many carers (i.e. break down into number of participants in these three categories). Introduction • "It will therefore be important to identify those at greatest risk of developing dementia following stroke in order to implement strategies to reduce risk." Not sure what these are and whether they need to be targeted to high risk people – or of they are general strategies re smoking, diet, exercise then could be done across the population? This is addressed to some extent in the discussion, but I feel needs to be raised in the rationale for the project.• "stroke-specific models predict dementia or cognitive impairment over a relatively short time period." Could the authors please give
--

more details about what time frame this involves (i.e. weeks or months?)

- What is the rationale for the 12 month follow up interview?

Method

- It was helpful to analyse and report carers and patients results separately.

- Why were some people see 12 months post initial appointment and some after 6 months and some both?

- How long were the interviews for each of the three groups?

Please give average length and range in minutes.

- The authors indicated on the Standards for Reporting Qualitative Research Checklist that reflexivity and author characteristics were not applicable to this study. However some of the language used indicate that understanding the perspectives of the researcher conducting the analysis may have been helpful (see comments below re Results section).

- Page 7 line 24 the authors state that clinician views have been reported elsewhere, although they then go on to present clinician views in this paper. It would be helpful to clarify that the clinician views are both presented elsewhere and within this paper.

Results

- “In total, 30 baseline interviews were conducted, analysed and compared including: 15 patient and carer interviews (see table 1) and 17 primary and secondary care clinician interviews”. This seems to be 32 baseline interviews, rather than the 30 stated? Please could you clarify this statement?

- “Eight stroke survivors and four carers agreed to a further follow-up interview six months later with nine interviews completed”. Again it is unclear why there were 9 not 12 follow up interviews. Please clarify.

- “how ignorance and not knowing about one’s risk would actually be more preferable” The term ‘ignorance’ suggest a pejorative stance from the author. It would have been helpful for the authors to reflect on their own stance and identify potential biases which might have affected their interpretation of the data. Please add a section on this.

- Participants raised useful thoughts around the benefits of being diagnosed, concerns about hypervigilance and falling into a sick role.

- There is a small range of participant quotes provided.

Discussion

- It remains unclear what interventions could be offered those identified at high risk. Need how likely these are to be helpful. This would seem to be a key point of the paper.

- There was no mention of capacity in relation to whether participants could make the decision to have the dementia screening. This would have been helpful especially when considering people with more severe stroke symptoms.

- It might have been helpful to reference other populations such as people with Downs syndrome who have guidance suggesting screening for dementia is useful.

- The need for useful interventions post dementia diagnosis is surely key in order for the assessments to be justifiable from both an ethical and resource perspective.

REVIEWER	Bogna Drozdowska University of Glasgow, United Kingdom
REVIEW RETURNED	22-Aug-2018

GENERAL COMMENTS	Thank you for the opportunity to review this manuscript. I found the topic of the study to be of high importance, and at the same time - it strikes me as one that is often neglected. Methods of assessing risk for development of a particular disease/negative outcome have become of considerable interest to many clinicians, researchers, and funding bodies worldwide. This also appears to be the case in the area of post-stroke cognitive function. This does not seem surprising, given our increasing understanding of the cognitive sequelae of stroke, and their importance to recovery and quality of life, as highlighted by stroke survivors themselves. However, developers of risk assessment tools tend to predominantly (if not solely) focus on the optimal selection of predictors, and statistical methods for model derivation and validation, ensuring high accuracy of predictions. Although undoubtedly these issues are essential, there seems to be limited consideration of the implications of using such a tool in clinical practice for patients and clinicians, how these assessments will be received by them, and how estimated risks should be communicated and acted upon. What adds to the uniqueness of this study, other than the topic itself, is the use of qualitative methods. This allows to obtain insights that would be difficult, or even impossible, to uncover through more commonly used quantitative methods. I would like to highlight, however, that my own experience of qualitative methods is very limited, which may be reflected in some of my concerns and suggestions. Please find below my comments on specific aspects of the manuscript: Major points: 1. In the introduction it is stated that “a history of stroke doubles the risk of incident dementia in older populations”. This is indeed the conclusion the authors of the referenced systematic review had reached, however I think more evidence around prevalence, from a wider range of studies, should be presented in this paragraph. The current phrasing of this fragment implies a high level of certainty, yet even in the systematic review it is indicated that actually only 5 out of 16 studies reported a doubling of risk of incident dementia following stroke, with others suggesting different estimates. The referenced review also did not include a meta-analysis, study quality was not formally assessed, and finally – as pointed out by Pendlebury and Rothwell, 2009 – observed prevalence of dementia is likely to depend on a number of different factors, including the context of a particular study, inclusion/exclusion criteria, etc. The same sentence then continues with: “a risk that is independent of demographic and cardiovascular risk factors and prestroke cognitive decline”. Once again, there are a range of studies providing evidence to the contrary, including the Pendlebury and Rothwell review, also referenced in this manuscript. Including reports in support of the latter view seems
---

particularly relevant here, as the paragraph concludes with: "It will therefore be important to identify those at greatest risk of developing dementia following stroke in order to implement strategies to reduce risk". If we assume that this risk is only associated with having a stroke, and in our interest group it had not been prevented, then it seems nothing can be done to improve future outcome, with apparently no other modifiable factors playing any role.

2. I would suggest clarifying the number of conducted interviews, as reported in the abstract and results section. It is reported that 30 baseline interviews were completed, yet 15 patient and carer interviews, and 17 clinician interviews, would indicate 32. Further, it is mentioned in the results section that 12 participants agreed to do a follow-up assessment, of which 9 were completed, yet in Table 1 in the "follow-up interview conducted" it says "yes" for 11 participants, with no information for C5. I think it would also be worth specifying why not all previously agreed follow-up interviews were conducted – e.g. were the patients lost to follow-up or changed their mind regarding participation?

3. I think it would be beneficial to provide some further information on patient, carer and clinician sampling in both the methods and results sections. Namely, it is mentioned that the participants were selected "purposively", is this to suggest they were not recruited consecutively? Were all stroke clinic patients over 60 with memory concerns informed of the study or only some, and were all those informed later contacted by the researcher? What was the recruitment period? On what basis were the clinicians chosen, apart from ensuring that different professional backgrounds were represented in the group? I think in this context it would also be useful to know how many of those invited to take part in the study ended up participating.

4. I think it would be important to include what information the participants were given regarding the risk assessment process, as this would be likely to influence their opinion. Were publications with references 9-12 the ones provided as examples? I assume there would have been a pre-written script or something similar for this purpose. Perhaps this could be included in a supplementary file?

5. Continuing from the previous point, readers and researchers/clinicians wanting to collect similar data, might be interested in the interview protocol. This would enable study replication, and potentially aid in understanding the results through providing additional context information. Once again, this could be provided as a supplementary document.

6. Based on the provided checklist (Supplementary Table 1), I have noticed that the following information appears to be missing:

- Rationales for chosen qualitative approach and method of data analysis (S5, S14); I think it would also be of interest to readers to highlight whether themes were identified in an inductive way (data driven) or a theoretical way
- Criteria for deciding when no further sampling was necessary (S8)
- Start and stop dates of data collection and analysis (also mentioned in point 3.; S10)

	7. I think it might be worth adding to the discussion two more relevant issues, related to the risk assessment tools themselves. Firstly, it is mentioned that these tools require data that are readily available in primary care, yet this is not always the case. For example, SIGNAL2 requires rather complex imaging assessments, with variables like volume of white matter hyperintensities and global cortical atrophy. These types of variables are likely to be difficult to obtain, even in a hospital setting. Secondly, many decisions and opinions around use of risk assessment tools are likely to rely on their accuracy. As far as I am aware, for prediction of dementia, existing tools have undergone either very limited external validation or none at all. Even if the tools were confirmed to have good performance in a wide range of different patient populations, accuracy cannot be expected to be perfect. Therefore, it is worth considering the implications of patients being misclassified in terms of risk groups. Minor points:  1. I would suggest revising the “participants” section in the abstract – with so much key information in brackets, the sentence becomes quite difficult to read. 2. Continuing from the above point, although overall I found the manuscript to be very well written, it includes many long and complex sentences, which tends to introduce some issues. Where possible, to support ease of understanding the text, I would suggest breaking down some of these sentences into shorter ones. In order cases, I think it would be advantageous to revise the text to ensure correct structure and punctuation – lack of the latter makes some passages particularly difficult to understand, and necessary to read the same sentence multiple times. For example: “However, potential barriers have been identified in previous studies such as system-related factors (lack of support, time constraints) and training in dementia, would need to be addressed”. Perhaps the sentence could be improved by adding a comma before “such” and the word “which” before “would”. “Training in communicating the risk assessment process particularly in the context of dementia would be required if this were to be implemented in clinical practice.” Perhaps here commas before “particularly” and after “dementia”. 3. In some instances “potential diagnosis of dementia” seems to be used interchangeably with “assessment of risk for dementia”. I think this should be avoided, as prognostic tools are not meant to diagnose a disease, yet identify those at high risk of developing it – being at high risk does not equal being diagnosed with the disease, which in fact may never develop.
--	---

REVIEWER	Nathan Davies University College London, UK
REVIEW RETURNED	28-Aug-2018

GENERAL COMMENTS	Thank you for this very interesting paper on post-stroke dementia risk. This a good qualitative paper, i only have a few minor suggestions.
---

	Abstract The authors mention baseline and follow up interviews at 12 months. This suggests it is part of a trial? If so this should be mentioned. If not what is meant by baseline? Introduction A nice background to introduce the study. My only comment would be the authors discuss the use of everyday use of risk models for cardiovascular and cerebrovascular diseases - why is this any different to dementia and why do their feelings need to be explored in relation to the use of one for dementia? A short point here would be good to make it clear to the reader the importance. Methods Suggest some re organisation of the methods - data collection is discussed first and talks about the participants included (clinicians, patients and family etc.) before we know who the participants were and inclusion/exclusion. Data collection should come after sampling. Discussion There is limited discussion of the barriers and facilitators in relation to patients and carers views. It may be worth a couple of sentence or two on how do the barriers and facilitators from patients here relate to the diagnosis of other conditions? For example genetic conditions and whether patients would rather not know. Confidence among practitioners is also discussed in the findings. I think this is really important as it is often assumed patients are the ones reluctant to discuss sensitive topics. This relates well with practitioners reluctance to engage in difficult conversations regarding advance planning etc. It would be good to see some comparison here, linking early stages of dementia and the difficult conversations and consultations to that of some of the later stages of dementia discussions and consultations.
--	--

VERSION 1 – AUTHOR RESPONSE

Reviewer Name: Dr Olivia Hewitt Institution and Country: The Oxford Institute of Clinical Psychology Training, University of Oxford, UK Please state any competing interests or state 'None declared': None declared	
Thank you for submitting this interesting paper for review. I hope you will find the following comments helpful. Formatting  The size and format of the text should be consistent throughout the paper. Abstract  It's not clear what 'using a risk model to predict post stroke dementia' actually means in this context. 	Thank you – this has now been corrected (not tracked changed) This has been changed to “risk assessment tool” to be consistent with the previous sentence This has been changed to “were conducted and analysed thematically”. Apologies for the

 • “Qualitative semi-structured interviews were conducted with thematic analysis.” Please change to clarify that the interviews were analysed (not conducted) using thematic analysis. How long post stroke were the interviews conducted? Were they conducted individually (one to one)? • Section on ‘Participants’ needs re wording as is not currently grammatically correct. • Thirty-nine (17 clinicians and 15 stroke patients and their carers at baseline) 17 plus 15 isn’t 39 – could you be explicit about how many patients are seen, how many staff and how many carers (i.e. break down into number of participants in these three categories). 	omission but this has been corrected and they were interviewed 6 and 12 months post-stroke. Details of whether they were conducted individually or not can be found in the main text due to word count limitations. We have altered the sentence structure to make it clearer to the reader Thank you for the opportunity to clarify. Not all interviews were conducted independently. Some patients and carers wanted to be interviewed together. Details have been added to the main text for clarity.
Introduction	
“It will therefore be important to identify those at greatest risk of developing dementia following stroke in order to implement strategies to reduce risk.” Not sure what these are and whether they need to be targeted to high risk people – or of they are general strategies re smoking, diet, exercise then could be done across the population? This is addressed to some extent in the discussion, but I feel needs to be raised in the rationale for the project.	Thank you for pointing this out – we have added a sentence to the introduction to raise this important point regarding strategies in whole populations but not specifically mentioned high risk groups.
“stroke-specific models predict dementia or cognitive impairment over a relatively short time period.” Could the authors please give more details about what time frame this involves (i.e. weeks or months?)	The time frame has been added to the text
What is the rationale for the 12 month follow up interview?	The 12 month follow-up interview was offered as a standalone interview time point to give participants the opportunity to delay contact with the research team to assist with the study if they didn’t feel that they were able to cope with participating sooner, but still wanted to help. Given that the majority of participants were followed up at the 12 month mark it also provided an opportunity to see how participants managed over a longer time period post-stroke.
It was helpful to analyse and report carers and patients results separately.	Each interview was analyzed individually; we grouped them into professionals (e.g. clinicians) and non-professionals (e.g. patients and carers) rather than creating an additional carer group. The point of interest was professional versus non-professional, rather than comparing carer and patient views.
Why were some people see 12 months post initial appointment and some after 6 months and some both?	Thank you for the opportunity to clarify. Clinicians were interviewed once. Patients and carers were offered the opportunity to be interviewed immediately after their stroke

	appointment and/or 12 months post-stroke (e.g. 6 months later). An interval of 6 months was considered sufficiently long enough for patients and carers to have recognized any new memory difficulties and considered the impact or compensatory strategies. The timing and frequency of interviews was dictated by the participants' themselves and the majority of participants opted for both time points.
How long were the interviews for each of the three groups? Please give average length and range in minutes	Thank you for allowing us to clarify. The interviews themselves covered a broad range of topics and as mentioned in the manuscript this paper only reports on the data for risk assessment tools. The length of the interviews that we could report would be the length of the entire interview and would not reflect the length of time participants talked about the subject of risk assessment so we have not included this in the paper.
The authors indicated on the Standards for Reporting Qualitative Research Checklist that reflexivity and author characteristics were not applicable to this study. However some of the language used indicate that understanding the perspectives of the researcher conducting the analysis may have been helpful (see comments below re Results section).	Thank you, we have added in the detail for EYHT's background i.e. he is a medical doctor to ensure readers understand the perspective of the researcher. This is also mentioned as a limitation of the study. To clarify the interviews were conducted by the EYHT who is a medical doctor with no previous known relationship with participants. At the time the patients were recruited via the stroke clinic. Due to the limited geographical nature of where the study was held, the patients may themselves have been patients at the GP practice of the interviewer. However he did not have any prior clinical relationship with the participants at the time of the interviews.
Page 7 line 24 the authors state that clinician views have been reported elsewhere, although they then go on to present clinician views in this paper. It would be helpful to clarify that the clinician views are both presented elsewhere and within this paper.	Thank you – we have added some additional text in the paper to clarify this. Due to the richness of the data and the different topics covered, we have reported these topics separately. The references have been included to signpost the reader to these other publications which came from the same interview process – please see references 18 and 19 (in publication).
Results	
“In total, 30 baseline interviews were conducted, analysed and compared including: 15 patient and carer interviews (see table 1) and 17 primary and secondary care clinician interviews”. This seems to be 32 baseline interviews, rather than the 30 stated? Please could you clarify this statement?	We have clarified this in the paper – essentially some interviews were conducted in pairs at both baseline and follow-up.

“Eight stroke survivors and four carers agreed to a further follow-up interview six months later with nine interviews completed”. Again it is unclear why there were 9 not 12 follow up interviews. Please clarify.	
“how ignorance and not knowing about one’s risk would actually be more preferable” The term ‘ignorance’ suggest a pejorative stance from the author. It would have been helpful for the authors to reflect on their own stance and identify potential biases which might have affected their interpretation of the data. Please add a section on this.	Thank you for highlighting this. This was clearly a badly phrased sentence. The term ignorance was used to try and explain the following quote rather than the author’s stance. We have removed the term ignorance and hope that the reviewer agrees that a separate section on the author’s stance is not needed as there is more background information already provided i.e. interviewer was a clinician. We have discussed this in the limitations i.e. some issues of having a clinician performing the interviews but we have not specifically discussed the stance of the interviewer as the data was eventually discussed by the research team as a whole.
Participants raised useful thoughts around the benefits of being diagnosed, concerns about hypervigilance and falling into a sick role. There is a small range of participant quotes provided.	Thank you for this comment. We feel that the quotes used in this manuscript are representative of the sample interviewed.
Discussion	
It remains unclear what interventions could be offered those identified at high risk. Need how likely these are to be helpful. This would seem to be a key point of the paper.	This paper is about risk assessment rather than interventions. Although we do certainly agree that the intervention component is important that is not the focus of this paper and ongoing research is needed which is mentioned in conclusions and future research sections.
There was no mention of capacity in relation to whether participants could make the decision to have the dementia screening. This would have been helpful especially when considering people with more severe stroke symptoms.	Thank you for this point – this is an important point and we have added a sentence with this in mind to clinical implications. It may also require additional extra research to understand how this can be achieved for people with language problems, at what point a lack of capacity itself indicates a memory problem that would make risk screening redundant
It might have been helpful to reference other populations such as people with Downs syndrome who have guidance suggesting screening for dementia is useful.	Thank you for this point which is important in the general context of screening for dementia certainly. However, our focus is on stroke-populations so have decided to focus the discussion on these at-risk individuals and the context of learning difficulties is very different.
The need for useful interventions post dementia diagnosis is surely key in order for the assessments to be justifiable from both an ethical and resource perspective.	Thank you for this point – this is certainly what the participants in this study have discussed.

Reviewer: 2 Reviewer Name: Bogna Drozdowska Institution and Country: University of Glasgow, United Kingdom Please state any competing interests or state 'None declared': None declared.	
Thank you for the opportunity to review this manuscript. I found the topic of the study to be of high importance, and at the same time - it strikes me as one that is often neglected. Methods of assessing risk for development of a particular disease/negative outcome have become of considerable interest to many clinicians, researchers, and funding bodies worldwide. This also appears to be the case in the area of post-stroke cognitive function. This does not seem surprising, given our increasing understanding of the cognitive sequelae of stroke, and their importance to recovery and quality of life, as highlighted by stroke survivors themselves. However, developers of risk assessment tools tend to predominantly (if not solely) focus on the optimal selection of predictors, and statistical methods for model derivation and validation, ensuring high accuracy of predictions. Although undoubtedly these issues are essential, there seems to be limited consideration of the implications of using such a tool in clinical practice for patients and clinicians, how these assessments will be received by them, and how estimated risks should be communicated and acted upon. What adds to the uniqueness of this study, other than the topic itself, is the use of qualitative methods. This allows to obtain insights that would be difficult, or even impossible, to uncover through more commonly used quantitative methods. I would like to highlight, however, that my own experience of qualitative methods is very limited, which may be reflected in some of my concerns and suggestions. Please find below my comments on specific aspects of the manuscript:	
Major points:	

In the introduction it is stated that “a history of stroke doubles the risk of incident dementia in older populations”. This is indeed the conclusion the authors of the referenced systematic review had reached, however I think more evidence around prevalence, from a wider range of studies, should be presented in this paragraph. The current phrasing of this fragment implies a high level of certainty, yet even in the systematic review it is indicated that actually only 5 out of 16 studies reported a doubling of risk of incident dementia following stroke, with others suggesting different estimates. The referenced review also did not include a metanalysis, study quality was not formally assessed, and finally – as pointed out by Pendlebury and Rothwell, 2009 – observed prevalence of dementia is likely to depend on a number of different factors, including the context of a particular study, inclusion/exclusion criteria, etc. The same sentence then continues with: “a risk that is independent of demographic and cardiovascular risk factors and prestroke cognitive decline”. Once again, there are a range of studies providing evidence to the contrary, including the Pendlebury and Rothwell review, also referenced in this manuscript. Including reports in support of the latter view seems particularly relevant here, as the paragraph concludes with: “It will therefore be important to identify those at greatest risk of developing dementia following stroke in order to implement strategies to reduce risk”. If we assume that this risk is only associated with having a stroke, and in our interest group it had not been prevented, then it seems nothing can be done to improve future outcome, with apparently no other modifiable factors playing any role.

Thank you for this important point – we have actually now removed this sentence and referenced a recent meta-analysis of dementia risk in stroke which included a greater number of pooled studies. It is important to note that although stroke increases one’s risk, it does not necessarily mean nothing can be done to prevent dementia as preventing recurrent stroke (which also contributes to increased risk of incident dementia) is also known to be important. The emphasis is also not just on prevention of dementia but also preparation for a potential diagnosis of dementia.

I would suggest clarifying the number of conducted interviews, as reported in the abstract and results section. It is reported that 30 baseline interviews were completed, yet 15 patient and carer interviews, and 17 clinician interviews, would indicate 32. Further, it is mentioned in the results section that 12 participants agreed to do a follow-up assessment, of which 9 were completed, yet in Table 1 in the “follow-up interview conducted” it says “yes” for 11 participants, with no information for C5. I think it would also be worth specifying why not all previously agreed follow-

Thank you – further details have been added to clarify these points including reasons why participants did not receive a follow-up interview.

up interviews were conducted – e.g. were the patients lost to follow-up or changed their mind regarding participation?	
I think it would be beneficial to provide some further information on patient, carer and clinician sampling in both the methods and results sections. Namely, it is mentioned that the participants were selected “purposively”, is this to suggest they were not recruited consecutively? Were all stroke clinic patients over 60 with memory concerns informed of the study or only some, and were all those informed later contacted by the researcher? What was the recruitment period? On what basis were the clinicians chosen, apart from ensuring that different professional backgrounds were represented in the group? I think in this context it would also be useful to know how many of those invited to take part in the study ended up participating.	Thank you for this point. The study had a range of inclusion and exclusion criteria (including over 60 and experiencing memory difficulties – see page 7 for inclusion criteria) which research nurses were asked to consider when initially approaching potential participants to ask if they might be interested in taking part in the study. We sought to recruit a range of patient experiences and a range of carers e.g (wife, husband, child). Only potential participants who met the inclusion criteria were invited to participate in the study by the stroke specialist nurse. If they agreed their contact details were then passed to the researcher who then contact them to explain the study in more detail. At this point some potential participants were not included as they were found not to be suitable or declined further involvement. We sought to employ purposive sampling to explore the range of patient and carer experiences. Recruitment took place over a protracted period of time (from April 2016 – January 2017) because we were interested in a specific population of stroke-survivors (over 60 with memory difficulties). Throughout the recruitment we reflected on the nature of the sample to ensure that our data included a range of patient and carer relationships. With reference to clinician sampling. We sought to include a range of different clinical backgrounds rather than demographics. Due to the variety of ways we recruited e.g. email contact and via networks for GP recruitment, it is not possible to determine with certainty those who declined.
I think it would be important to include what information the participants were given regarding the risk assessment process, as this would be likely to influence their opinion. Were publications with references 9-12 the ones provided as examples? I assume there would have been a pre-written script or something similar for this purpose. Perhaps this could be included in a supplementary file?	There was no pre-written script for this purpose but a few examples of risk assessment tools for dementia were shown to participants – these were not the stroke specific risk assessment tools but were scores where the layout, variables etc would be understood by all participants including patients and carers. The purpose of the study was to understand whether this concept after stroke would be acceptable if developed, and how it should be used.
Based on the provided checklist (Supplementary Table 1), I have noticed that the following information appears to be missing:	

 • Rationales for chosen qualitative approach and method of data analysis (S5, S14); I think it would also be of interest to readers to highlight whether themes were identified in an inductive way (data driven) or a theoretical way • Criteria for deciding when no further sampling was necessary (S8) • Start and stop dates of data collection and analysis (also mentioned in point 3.; S10) 	 - The qualitative approach (S5) is constant comparative analysis which is in the text. The method of data analysis was thematic which is also in the text i.e. reflecting the data itself - Criteria for sampling saturation has been added. - Data collection start and stop dates are in the text. It is difficult to specify exactly when analysis stops as analysis also involved the wider research team, and indeed some analysis is still on-going related to other aspects of the broader study.
I think it might be worth adding to the discussion two more relevant issues, related to the risk assessment tools themselves. Firstly, it is mentioned that these tools require data that are readily available in primary care, yet this is not always the case. For example, SIGNAL2 requires rather complex imaging assessments, with variables like volume of white matter hyperintensities and global cortical atrophy. These types of variables are likely to be difficult to obtain, even in a hospital setting. Secondly, many decisions and opinions around use of risk assessment tools are likely to rely on their accuracy. As far as I am aware, for prediction of dementia, existing tools have undergone either very limited external validation or none at all. Even if the tools were confirmed to have good performance in a wide range of different patient populations, accuracy cannot be expected to be perfect. Therefore, it is worth considering the implications of patients being misclassified in terms of risk groups.	Thank you for raising this point. The main purpose of the study was initial acceptability and utility of a risk assessment concept? We did not specifically explore about false positive and negative outcomes from a risk assessment – which would require additional examination informed by observational data from an assessment tool in use. We have however added the following paragraph to try and address these two points: “Further, some models, particularly those developed in stroke populations¹¹ may also include variables such as complex imaging data, which will only be available in secondary care and may be difficult to obtain even in specialist settings. If risk assessment were to be conducted in primary care, then the risk assessment models utilising data which can be accessed in primary care, needs to be externally validated in stroke populations to assess their accuracy”.
Minor points:	
I would suggest revising the “participants” section in the abstract – with so much key information in brackets, the sentence becomes quite difficult to read.	We have revised this
Continuing from the above point, although overall I found the manuscript to be very well written, it includes many long and complex sentences, which tends to introduce some issues. Where possible, to support ease of understanding the text, I would suggest breaking down some of these sentences into shorter ones. In order cases, I think it would be advantageous to revise the text to ensure correct structure and punctuation – lack of the	We have been through the manuscript and made some changes to help with the flow.

latter makes some passages particularly difficult to understand, and necessary to read the same sentence multiple times. For example: “However, potential barriers have been identified in previous studies such as system-related factors (lack of support, time constraints) and training in dementia, would need to be addressed”. Perhaps the sentence could be improved by adding a comma before “such” and the word “which” before “would”. “Training in communicating the risk assessment process particularly in the context of dementia would be required if this were to be implemented in clinical practice.” Perhaps here commas before “particularly” and after “dementia”.	
In some instances “potential diagnosis of dementia” seems to be used interchangeably with “assessment of risk for dementia”. I think this should be avoided, as prognostic tools are not meant to diagnose a disease, yet identify those at high risk of developing it – being at high risk does not equal being diagnosed with the disease, which in fact may never develop.	Thank you for this important point. We appreciate the difference between the two terms. We think the key point from the interviewees perspective was that an assessment outcome of being high risk could lead to a potential diagnosis of dementia. This potential for a diagnosis would then generate anxiety which is brought about by the risk assessment process. It would be hard to talk about risk without talking about potential for a diagnoses particularly from the patient/carer perspective. The term potential diagnosis has been kept to mirror the views of the participants.
Reviewer: 3 Reviewer Name: Nathan Davies Institution and Country: University College London, UK Please state any competing interests or state ‘None declared’: none declared	
Abstract The authors mention baseline and follow up interviews at 12 months. This suggests it is part of a trial? If so this should be mentioned. If not what is meant by baseline?	Thank you – this was not part of a trial and the definition of baseline and follow-up have been added.
Introduction A nice background to introduce the study. My only comment would be the authors discuss the use of everyday use of risk models for cardiovascular and cerebrovascular diseases - why is this any different to dementia and why do their feelings need to be explored in relation to the use of one for dementia? A short point here would be good to make it clear to the reader the importance.	We have added the following: “... particularly given the stigma surrounding the diagnosis and perceived limited interventions and increased awareness of cognitive difficulties that patients and carers may have following stroke.”

Methods Suggest some re organisation of the methods - data collection is discussed first and talks about the participants included (clinicians, patients and family etc.) before we know who the participants were and inclusion/exclusion. Data collection should come after sampling.	Thank you for this important point – we have moved the sections as suggested
Discussion There is limited discussion of the barriers and facilitators in relation to patients and carers views. It may be worth a couple of sentence or two on how do the barriers and facilitators from patients here relate to the diagnosis of other conditions? For example genetic conditions and whether patients would rather not know.	We have added the following paragraph to discuss the facilitator “preparation” further. We feel that genetic conditions represent a very different population e.g. possibly a younger population and so would be difficult to compare “In the context of the dementia diagnostic journey, transition from living with an undiagnosed memory problem to being diagnosed with a dementia illness is underpinned by uncertainty³⁹. Although risk assessment does not provide any certainty about development of a dementia illness, the discussions and objective evaluation using the tools may help the individual’s process their current condition and assist in the preparation for a potential diagnosis of dementia. Preparation for memory problems was mentioned by participants in this study as a facilitator for risk assessment”.
Confidence among practitioners is also discussed in the findings. I think this is really important as it is often assumed patients are the ones reluctant to discuss sensitive topics. This relates well with practitioners reluctance to engage in difficult conversations regarding advance planning etc. It would be good to see some comparison here, linking early stages of dementia and the difficult conversations and consultations to that of some of the later stages of dementia discussions and consultations.	We have included the following in clinical implications: “It is also important to note that GPs find communicating the diagnosis of dementia difficult⁴¹. Although risk assessment is not providing a diagnosis of dementia, careful consideration will be required in training health professionals in communicating the concept of risk for a disease such as dementia.”

VERSION 2 – REVIEW

REVIEWER	Dr Olivia Hewitt The Oxford Institute of Clinical Psychology Training and Research The Oxford Centre for Psychological Health Isis Education Centre Warneford Hospital Oxford OX3 7JX
REVIEW RETURNED	12-Nov-2018

GENERAL COMMENTS	Thank you for sending a revised version of this paper. I had a couple of further queries: “A recent meta-analysis found that both prevalent and incident strokes are strong independent risk factors for dementia” Do the authors mean that both the prevalence rates and incidence rates of strokes are risk factors for dementia? This needs rephrasing. Are there clear inclusion and exclusion criteria for participants in each group?
--

REVIEWER	Bogna Drozdowska University of Glasgow
REVIEW RETURNED	26-Nov-2018

GENERAL COMMENTS	Thank you for the opportunity to re-review this paper. I think that the manuscript has undergone a number of advantageous changes, I would however still like to suggest a few more adjustments for your consideration:  1. Regarding a previous comment on including more details of recruitment and the study participants, the response was: “The study had a range of inclusion and exclusion criteria (including over 60 and experiencing memory difficulties – see page 7 for inclusion criteria) which research nurses were asked to consider when initially approaching potential participants to ask if they might be interested in taking part in the study.” The word “range” suggests that there were more criteria applied than just the two mentioned in the text (over 60 and subjective memory compliant. If this is the case, I would recommend adding all the inclusion/exclusion criteria to the methods section, so that this matter is transparent. Further, in the same response it is explained that after contacting potential participants some “were not included as they were found not to be suitable”. I think it is relevant to mention on what basis this judgment was made or provide examples of such cases. 2. In relation to a previous comment (Based on the provided checklist (Supplementary Table 1), I have noticed that the following information appears to be missing: Rationales for chosen qualitative approach and method of data analysis (S5, S14); I think it would also be of interest to readers to highlight whether themes were identified in an inductive way (data driven) or a theoretical way...), the response was “The qualitative approach (S5) is constant comparative analysis which is in the text. The method of data analysis was thematic which is also in the text i.e. reflecting the data itself”. Firstly, there remains no explanation of the rationale for choosing constant comparative analysis. Secondly, thematic analysis can be either inductive or theoretical/deductive – the type of method used has not been clearly stated in the text, although the response seems to suggest that an inductive approach was applied. 3. The apparent lack of consistency between the number of participants and interviews has now been clarified in the main body of the manuscript, however in the abstract it seems that the reported numbers are still likely to confuse readers. Perhaps a short comment can be added to say that some interviews were conducted in pairs, to explain this perceived mismatch.
--

	4. Page 8, lines 53-58: I think it would be useful to reference here the tools that were provided as examples for the participants. 5. Given that only some of the data from the interviews are presented here, and some topics seem less prominent in the participants' responses than others (e.g. on variable use), I think the interview process would be more transparent to readers if the interview guide (the part relevant to this specific paper), with a list of questions and topics that needed to be covered, was presented in the supplementary materials. 6. Page 11, from line 42: "For some stroke-survivors it did not matter who was performing the risk assessment for dementia or where it was undertaken. What was important was that the diagnosis was reached at the right time: "I wouldn't say it matters, as long as it's diagnosed at the right time." (P5, male stroke-survivor) To enable this, when discussing who should perform the risk assessment, carer participants felt that primary care and the community were regarded as being optimal, because of the existing GP-patient relationship. This is because the GP has an overall view of the individual's care". The phrase "to enable this" suggests that the latter thought is in line with/a continuation of the former, however I think these two opinions are in fact contradicting. One person emphasises that time is a priority, with the healthcare setting being less relevant, while the other emphasises that the relationship with the clinician is most important and advocates specifically for a GP to carry out the assessment. 7. Page 22, lines 10-15: Perhaps a reference could be added to support this statement.
--	--

REVIEWER	Nathan Davies University College London, UK
REVIEW RETURNED	26-Nov-2018

GENERAL COMMENTS	I think this is a great article and happy with all authors responses and changes.
---

VERSION 2 – AUTHOR RESPONSE

Reviewer 1	
Thank you for sending a revised version of this paper. I had a couple of further queries: "A recent meta-analysis found that both prevalent and incident strokes are strong independent risk factors for dementia" Do the authors mean that both the prevalence rates and incidence rates of strokes are risk factors for dementia? This needs rephrasing.	Thank you for the opportunity to clarify. We have removed prevalent and incident and simply highlighted that stroke itself is a strong independent risk factor for dementia.
Are there clear inclusion and exclusion criteria for participants in each group?	Criteria for inclusion into the study can be found on page 7. We have also included the fact that participants must be able to communicate effectively in English in order to participate in this qualitative study.

Reviewer: 2	
Thank you for the opportunity to re-review this paper. I think that the manuscript has undergone a number of advantageous changes, I would however still like to suggest a few more adjustments for your consideration: 1. Regarding a previous comment on including more details of recruitment and the study participants, the response was: “The study had a range of inclusion and exclusion criteria (including over 60 and experiencing memory difficulties – see page 7 for inclusion criteria) which research nurses were asked to consider when initially approaching potential participants to ask if they might be interested in taking part in the study.” The word “range” suggests that there were more criteria applied than just the two mentioned in the text (over 60 and subjective memory compliant. If this is the case, I would recommend adding all the inclusion/exclusion criteria to the methods section, so that this matter is transparent. Further, in the same response it is explained that after contacting potential participants some “were not included as they were found not to be suitable”. I think it is relevant to mention on what basis this judgment was made or provide examples of such cases.	Thank you for allowing us to clarify – our apologies if there was any confusion in the use of the word “range” – in terms of the study, the inclusion criteria remain as written in the manuscript. Eligibility again depended on the criteria that were set for the study and approved by NHS Ethics review.
2. In relation to a previous comment (Based on the provided checklist (Supplementary Table 1), I have noticed that the following information appears to be missing: Rationales for chosen qualitative approach and method of data analysis (S5, S14); I think it would also be of interest to readers to highlight whether themes were identified in an inductive way (data driven) or a theoretical way...), the response was “The qualitative approach (S5) is constant comparative analysis which is in the text. The method of data analysis was thematic which is also in the text i.e. reflecting the data itself”. Firstly, there remains no explanation of the rationale for choosing constant comparative analysis. Secondly, thematic analysis can be either inductive or theoretical/deductive – the type of method used has not been clearly stated in the text, although the response seems to suggest that an inductive approach was applied.	We have added a paragraph to clarify this point on page 9.
3. The apparent lack of consistency between the number of participants and interviews has now been clarified in the main body of the manuscript, however in the abstract it seems that the reported numbers are still likely to confuse readers.	This has been included in the abstract

Perhaps a short comment can be added to say that some interviews were conducted in pairs, to explain this perceived mismatch.	
4. Page 8, lines 53-58: I think it would be useful to reference here the tools that were provided as examples for the participants.	As these tools were not stroke-specific risk assessments tools but merely used to demonstrate the process of risk assessment, we don't believe it would add any value to the manuscript by having this referenced. Reference 10 provides a systematic review of some examples of dementia specific risk assessment tools which might be helpful for the reader.
5. Given that only some of the data from the interviews are presented here, and some topics seem less prominent in the participants' responses than others (e.g. on variable use), I think the interview process would be more transparent to readers if the interview guide (the part relevant to this specific paper), with a list of questions and topics that needed to be covered, was presented in the supplementary materials.	Thank you for the opportunity to respond. We have actually included the topics covered in the actual manuscript text. For further clarity we have further elaborated on this text (see page 8)
6. Page 11, from line 42: "For some stroke-survivors it did not matter who was performing the risk assessment for dementia or where it was undertaken. What was important was that the diagnosis was reached at the right time: "I wouldn't say it matters, as long as it's diagnosed at the right time." (P5, male stroke-survivor) To enable this, when discussing who should perform the risk assessment, carer participants felt that primary care and the community were regarded as being optimal, because of the existing GP-patient relationship. This is because the GP has an overall view of the individual's care". The phrase "to enable this" suggests that the latter thought is in line with/a continuation of the former, however I think these two opinions are in fact contradicting. One person emphasises that time is a priority, with the healthcare setting being less relevant, while the other emphasises that the relationship with the clinician is most important and advocates specifically for a GP to carry out the assessment.	Thank you for highlighting this, we have removed the phrase "to enable this" to help clear this point up.
7. Page 22, lines 10-15: Perhaps a reference could be added to support this statement.	Presumably this is referring to the section on inflammation. We have amended the sentence and added a reference on page 22 to clarify this point if this is what the reviewer was referring to.
Reviewer: 3	
I think this is a great article and happy with all authors responses and changes.	Thank you

VERSION 3 - REVIEW

REVIEWER	Bogna Drozdowska University of Glasgow, United Kingdom
REVIEW RETURNED	07-Feb-2019

GENERAL COMMENTS	Thank you for replying to all comments and introducing suggested adjustments. I am happy with all of the authors' responses. I think the paper reads very well and it should be of great interest to a wide range of BMJ Open readers. I have only noticed a few very minor errors in the text: Page 7, lines 14-16: I think some words may have been omitted in the 2nd part of the sentence, following "i.e." Page 10, line 8: instead of "data software handing (no 'l') package" I think it should be "data handling software package" Page 19, line 55: there is no "be" after "would", and I think the sentence would read better with a coma before "particularly"
---